# The SlymiR394-ZDS Module Enhances the Antioxidant Capacity of Tomato by Promoting Lycopene Synthesis

**DOI:** 10.3390/biom15060857

**Published:** 2025-06-12

**Authors:** Qiuyu Zhao, Li Zhao, Jiuzhi Shi, Xiaoxiao Chen, Zhenchao Yang, Yongjun Wu

**Affiliations:** 1College of Life Sciences, Northwest A&F University, Yangling 712100, China; zhaoqy@nwafu.edu.cn (Q.Z.); zhaoli777@nwafu.edu.cn (L.Z.); shijiuzhi@nwafu.edu.cn (J.S.); 2020010302@nwafu.edu.cn (X.C.); 2College of Horticulture, Northwest A&F University, Yangling 712100, China

**Keywords:** lycopene, miR394, *ZDS*, antioxidant capacity, tomato

## Abstract

Lycopene, a natural pigment, is valuable for human health because of its strong antioxidant capacity. However, studies on the involvement of tomato miR394 in the regulation of lycopene have not been reported. The aim of this study was to reveal the molecular mechanism by which miR394 regulates lycopene synthesis by targeting ζ-carotene dehydrogenase (*ZDS*). The miR394-silenced transgenic tomato plants were constructed by short tandem target mimicry (STTM) technology, and the association between lycopene content and antioxidant capacity was analyzed by combining qRT-PCR, UV spectrophotometry, and a free radical scavenging assay. The targeting relationship between miR394 and *ZDS* was verified using a subcellular localization assay. The results showed that the silencing of miR394 significantly upregulated the expression of the ZDS gene and promoted lycopene accumulation. The antioxidant enzyme activities of STTM394 transgenic plants were significantly enhanced, and the free radical scavenging ability was obviously improved. Subcellular localization experiments confirmed that miR394 directly inhibited the chloroplast expression of *ZDS*. In conclusion, this study reveals for the first time that the miR394-ZDS module enhances the antioxidant capacity by regulating lycopene metabolism, which provides a new target for themolecular breeding of highly nutritious tomatoes.

## 1. Introduction

Lycopene is a natural pigment. It is a type of carotenoid that gives vegetables and fruits their red color and bright appearance. Lycopene is also a biologically active constituent found mainly in tomatoes, and its molecule consists of polyalkene chains in an open ring structure containing 11 conjugated and 2 non-conjugated double bonds, with no aromatic ring at the end [1]. This structure gives the molecule a high antioxidant capacity, significantly higher than that of other carotenoids, being more than twice that of β-carotene and about 10 times that of α-tocopherol [2]. By scavenging free radicals, it protects cells from the oxidation of its components and has a strong antioxidant capacity [3]. Therefore, lycopene has many beneficial effects, including anti-inflammatory effects and cardioprotective effects, and it reduces cardiovascular disease risk [4,5,6]. In addition, lycopene is believed to treat certain cancers and human degenerative diseases [7,8]. Studies have shown that lycopene can prevent tumor promotion and progression by modulating fundamental signaling pathways activated by tumor promoters, inflammatory cytokines, and growth factors. By modulating these pathways, lycopene exerts a protective effect, thus contributing to the prevention and control of carcinogenesis [9,10]. Therefore, it is crucial to increase the lycopene content in tomatoes.

During tomato fruit development, dramatic color changes occur. In the early stages, the fruit is green, containing carotenoids essentially identical to those in green leaves. These are primarily composed of β-carotene, violaxanthin, and lutein. At the “breaker” stage of lycopene maturation, lycopene concentrations increase nearly 500-fold in ripening fruits [11]. During this process, the transcription of the *PSY* (encoding phytoene synthase) and *PDS* (phytoene desaturase) genes is upregulated [12,13,14]. Mature tomato fruits activate the carotenoid metabolic pathway by regulating the expression of key genes such as *PSY1*, *PDS*, and *ZDS*, leading to the massive accumulation of lycopene-dominated carotenoids [15,16]. These key enzymes and genes regulate lycopene synthesis in plants.

MicroRNAs (miRNAs) are a class of endogenous small-molecule non-coding RNAs some 20–25 nucleotides in length. They negatively regulate the expression of protein-coding genes through the mechanism of target mRNA cleavage or translational repression at the post-transcriptional level and play a central regulatory function in plant growth and development and metabolic pathways [17]. The mechanism of miRNA involvement in carotenoid synthesis has been studied in plants. In sweet orange, the target prediction of 60 differential miRNAs indicated that two key genes for carotenoid biosynthesis, geranylgeranyl pyrophosphate synthase (GGPS) and lycopene β-cyclase (LYCb), were under miRNA regulation [18]. In tomato, the overexpression of miR160a improved chlorophyll degradation and increased lycopene content [19]. The processing of artificial microRNA (amiRNA) stem–loop precursors to produce mature amiRNAs successfully targeted PSY mRNAs and reduced their levels in the alga Phyllostachys edulis, and carotenoid levels were also found to be reduced in the amiRNA knockout lines of Phyllostachys edulis [20]. In yellow peach pericarp, miR167d, miR394a, mi67585, and mdm-miR858 control carotenoid synthesis by targeting the genes *PSY2*, *ZDS1/2*, *CHYB*, *ZEP*, *VDE*, and *NCED1/3*, as well as the TFs *PpARF1*, *PpNAC1*, and *PpWRKY2* [21].

MiR394 is one of the many miRNA family members discovered whose precursor sequences vary across plants but where all contain the same highly conserved mature fragment [22]. Studies have shown that miR394 is involved in regulating morphological development in plants. For example, oilseed rape plants overexpressing miR394 exhibited features of delayed flowering and increased pod and seed size [23]. Conversely, rice, which repressed miR394 expression, exhibited a morphology with increased leaf angle, which was attributed to the fact that miR394 regulates the expansion and elongation of the proximal thin-walled cells by targeting LEAF INCLINATION 4, leading to leaf angle enlargement [24]. miR394 also plays an important role in plant resistance to abiotic stresses. For example, in the *Arabidopsis* low-temperature experiment, it was found that *Arabidopsis* plants overexpressing miR394 exhibited enhanced low-temperature tolerance [25], whereas in high-temperature environments, the expression of miR394 was significantly reduced in *Betula platyphylla* [26]. miR394 was found to negatively regulate salt stress in *Arabidopsis thaliana* in the *Arabidopsis* salt stress experiment. Plants showing miR394 overexpression and *LCR* mutant plants were more sensitive to salt stress, while plants showing *LCR* overexpression showed salt tolerance [27].

Tomato is one of the major vegetable crops grown in agricultural facilities around the world, and tomato and its processed products are commonly consumed. They are thus a major route of lycopene intake [28,29,30]. Several studies have been conducted on the role of miR394 in regulating plant development as well as resistance to abiotic stresses. Although miRNA-mediated regulatory networks of lycopene metabolism have been revealed in several species, it is not clear whether miR394 can regulate lycopene synthesis. Therefore, the aim of this study was to analyze the molecular mechanism by which tomato miR394 regulates the lycopene synthesis pathway. In this study, miR394 was found to be a key negative regulator of the lycopene synthesis pathway in tomato, and it was directly involved in the biosynthesis of lycopene by binding to *ZDS*, a key regulator of lycopene. It was found that the silencing of tomato miR394 by STTM technology promoted the expression of *ZDS*, which resulted in the accumulation of lycopene and improved the antioxidant capacity of tomato. These results reveal the mechanism by which the miR394-ZDS module enhances the antioxidant capacity of tomato by regulating the lycopene metabolic pathway, filling a gap in the metabolic regulatory network.

## 2. Materials and Methods

### 2.1. Plant Material and Growing Conditions

Four different varieties of tomato—“Micro Tom”, “red warbler”, “canary” and “dwarf pink bird”—were used in this study. Micro Tom was purchased from the Ball Horticultural Company, and the other varieties were purchased from Liyang Qiqu Partner Breeding Co. (LiYang, China).

Tomatoes were planted in soil–vermiculite 3:1 pots with a height of about 15 cm and a base diameter of about 18.5 cm at one plant per pot, and incubated using a light incubator at a temperature of 20 °C, with a light and dark time of 16/8 h and a light intensity of 180 μmol·m^−2^·s^−1^. Tomato fruits with relatively uniform growth were selected at the green, firm, and finished ripening stages.

### 2.2. RNA Extraction and Real-Time Quantitative PCR (qRT-PCR) Analysis

Total RNA was extracted using TRNzol Universal Total RNA Isolation Kit (Tiangen, Beijing, China). Subsequently, cDNA was synthesized from RNA using the Evo M-MLV RT Kit with gDNA Clean for qPCR II from Accurate Biotechnology (Changsha, China). Quantitative real-time PCR (qRT-PCR) was performed using 2×Universal SYBR Green Fast qPCR Mix (ABclonal, Wuhan, China) in the CFX96 real-time system (BioRad, Munich, Germany). U6 is the internal reference gene of miRNA, Actin is the internal reference gene of mRNA, and each gene has 3 biological repeats. The relative gene expression was calculated using 2^−ΔΔCt^. Please refer to Table 1 for primer sequences.

### 2.3. Determination of Red–Green Ratio of Tomato Fruits

The red–green ratio of fruit was determined by computer vision and image processing. We used the camera to take pictures of the fruit, process the pictures through ImageJ v1.8.0 software, separate the color layers, measure the red–green color, perform calculations to obtain the red–green ratio per unit area (R_mean/G_mean), and use the origin 2021 software to make graphs.

### 2.4. Extraction and Content Determination of Lycopene

Referring to the method of Akpolat H et al. [31], 400 mg of tomato samples was taken, hexane/anhydrous ethanol/acetone (2:1:1, *v*/*v*/*v*) was used as the extraction solvent, and catechol was added to prevent decomposition. The sample was vortexed for 1 min and then left to stand for 20 min under the protection of light. Then, 1 mL of deionized water was added to promote the separation of the phases, and the sample was further protected from light for 10 min. The absorbance was measured by a UV spectrophotometer at 503 nm and the concentration was calculated using the extinction coefficient of 172,000 M^−1^cm^−1^. Each sample was analyzed three times in parallel, and the content was expressed as mg/kg fresh weight.

### 2.5. Measurement of Antioxidant Enzyme Activities

Referring to the method of Shah et al. [32], known as the SOD Extraction and Activity Assay, tomato fruits (0.3 g) were homogenized in a pre-cooled extraction buffer (50 mmol/L phosphate buffer containing 1% PVP) using ice-bath grinding, adjusted to a final volume of 10 mL, and centrifuged at 1000 rpm (4 °C, 15 min). The reaction system contained NBT, EDTA, riboflavin, methionine (Met), and crude enzyme extract. Dark control tubes (enzyme solution replaced and shielded from light) were established. Reactions were illuminated at 100 μmol·m^−2^·s^−1^ for 15–20 min, and absorbance was measured at 560 nm. SOD activity was calculated according to the formula. POD Extraction and Activity Assay: Tomato fruits (0.3 g) were homogenized in a phosphate buffer via ice-bath grinding, adjusted to 5 mL, and centrifuged to collect supernatant. The reaction system contained H_2_O_2_, guaiacol, and enzyme extract. After 15 min incubation at 37 °C, reactions were terminated by ice bath and trichloroacetic acid addition. Absorbance at 470 nm was measured, with activity calculated based on ΔA470 per unit time. CAT Extraction and Activity Assay: Samples (0.3 g) were homogenized and centrifuged as above. The A2 mL aliquot was inactivated by boiling bath water as a control. Reactions containing Tris-HCl, H_2_O_2_, and enzyme extract were dynamically monitored at 240 nm (every 30 s for 3 min, 25 °C). Activity was determined via ΔA240 and the H_2_O_2_ decomposition rate.

### 2.6. Measurement of Free Radical Scavenging Capacity of DPPH and ABTS

DPPH and ABTS radical scavenging capacities were determined according to the method of Re et al. [33]. Tomato fruits were oven-dried and ground into powder. Then, 0.05 g of powder was mixed with 10 mL of absolute ethanol (5 mg/mL) for 65 °C water bath extraction. This was followed by centrifugation to collect supernatant.

DPPH assay: We dissolved 0.02 g of DPPH in 72 mL of ethanol. The 1 mL DPPH solution + 0.5 mL ethanol + sample solution (added until color fading). Sample gradients (based on maximum fading volume) were tested with triplicates. Scavenging rate (%) = [(A_0_ − A)/A_0_] × 100. 

ABTS assay: We dissolved 13.4 mg of K_2_S_2_O_8_ + 38.4 mg ABTS in 10 mL of H_2_O. The solution was mixed 1:1 (*v*/*v*), dark-incubated for 12 h, and diluted with phosphate buffer (pH 7.4) to A_734_ = 0.7. Sample preparation used the same method as DPPH. Standard (0.01–0.08 mg/mL) and sample gradients were measured. Each sample is repeated three times. Scavenging rate (%) = [(A_0_ − A)/A_0_] × 100.

### 2.7. Plasmid Construction

The precursor sequence of tomato miR394 was downloaded from the miRBase database (https://www.mirbase.org) (accessed on 24 June 2023), and the vector pOT2-miR394 was obtained by ligating pre-miR394 into the pOT2 vector through the *Kpn* I and *Bam*H I cleavage sites.

The sequence of tomato *ZDS* (NC_015438.3) was obtained from the NCBI database (https://www.ncbi.nlm.nih.gov/) (accessed on 24 June 2023), and *ZDS* was ligated into the pTF486-GFP vector through *Sal* I and *Bam*H I enzymatic sites to obtain pTF486-ZDS-GFP.

Referring to the short tandem target mimic (STTM) technology developed by Tang et al. [34], miR394 primers with a STTM stem–loop structure and *Swa* I cleavage sites at both ends were synthesized by PAGE, and miR394 fragments in pOT2-Poly-cis were replaced with those of the designed miR394 fragments through one-step inverse PCR with the help of the *Swa* I enzyme and T4 DNA Ligase V2 (Monad). Using one-step inverse PCR, the Poly-cis fragment of pOT2-Poly-cis was replaced by the designed miR394 fragment, which was cleaved and ligated by *Swa* I enzyme and T4 DNA Ligase V2 to obtain pOT2-STTM-miR394. pOT2-STTM-miR394 was obtained by PCR by removing the replication initiation site and introducing the *Pac* I cleavage site into the vector pOT2-STTM-miR394. The *Pac* I cleavage site, pOT2-STTM-miR394, and pFGC5941 were digested by *Pac* I and ligated with T4 DNA Ligase V2 to obtain pFGC5941-STTM-miR394. Please refer to Table 1 for primer sequences. All enzymes were purchased from Accurate Biology (Changsha, China).

### 2.8. Interaction Prediction and Characterization of miR394 and ZDS

We used the target website (https://www.zhaolab.org/psRNATarget/) (accessed on 24 June 2023) to predict the miR394-targeted lycopene biosynthetic pathway of the important enzyme gene [35].

Referring to the method of Jiang et al. to verify the interactions between miR394 and *ZDS* [36], experiments were performed using two plasmids, pOT2 and pTF486-GFP. The plasmids constructed were transfected with Agrobacterium. Single colonies with positive readings were picked. The corresponding antibiotics and rifampicin were added, respectively, and were subjected to the conditions of 28 °C and 220 rpm for amplification. The bulk-amplified bacterial solution was centrifuged at 6000 rpm for 5 min. The supernatant was discarded, and the bacterial samples were diluted to OD600 = 0.5 with 10 mM MgCl_2_, and then mixed 1:1 according to the following sequences: (1) pOT2-miR394 + pTF486-ZDS-GFP, (2) pOT2-miR394 + pTF486-GFP, (3) pOT2 + pTF486-ZDS-GFP, and (4) pOT2 + pTF486-GFP. After mixing the bacterial solution, we selected the four-leaf tobacco with healthy growth for 3–4 weeks for the injection of mixtures (1)–(4), during which the tobacco was pinched to ensure that the four pieces of tobacco at the base grew well. When injecting, a 1 mL syringe without a needle should be taken to inject the back of the tobacco, and the injection site should be marked with a marker. After the injection was completed, the tobacco was placed in an incubator with light/dark conditions of 16/8 h for 2–4 days. The lower epidermis of the tobacco in the injection area was torn for preparation and observation. The fluorescence display site and brightness were observed under a laser confocal microscope, and the fluorescence intensity was measured using Image J. The fluorescence intensity of the injected area was measured using a laser confocal microscope. In addition, the RNA of tobacco leaves in the injection area was extracted for reverse transcription and qPCR to detect the expression of GFP. Please refer to Table 1 for primer sequences.

### 2.9. Data Processing

All measurements were conducted in triplicate and expressed as mean values. Data were analyzed using IBM SPSS 27.0 and Microsoft Excel 2021. We used different letters to indicate the significance of the difference. The letters themselves do not represent fixed, independent meanings; they represent statistical “significance groupings”, where if two bars have the same letter, it means that the difference between them is not statistically significant (*p* > 0.05), and if two bars have different letters it means that the difference between them is statistically significant (*p* < 0.05). Letter order does not reflect data size. Figures were generated using Graphpad8 software.

## 3. Results

### 3.1. Lycopene Content in Tomato Cultivars with Distinct Fruit Colors

We photographed four different varieties of tomato fruits with a Canon EOS 5DsR camera (Canon Co., Beijing, China) to observe the differences in fruit color. As can be seen in Figure 1, there was no significant difference in the color of tomato fruits of the four varieties at the mature green stage, and as ripening increased, the fruit color at the turning stage began to show differences, and at the red and ripening stage, Micro Tom and dwarf pink bird appeared bright red. A cross-sectional view was obtained by cutting the fruits across the grain, and a clear difference in color could be seen (Figure 1A). We processed the images, separated the color layers, measured the red-to-green ratio, and calculated the red-to-green ratio per unit area (R_mean/G_mean). The results were consistent with the color variations we observed, with higher red-to-green ratios for Micro Tom and Pink Bird at the red and ripening stage suggesting that the fruits had a more vibrant red color (Figure 1B).

The increased synthesis of lycopene leads to the formation of a red color, and lycopene concentration increases with tomato ripening [37]. Thus, in order to more precisely characterize the differences in fruit pigmentation among the four varieties of tomato, we determined the content of lycopene in different varieties of tomato at the mature green stage, turning stage, and red and ripening stage. Different tomato varieties were basically free of lycopene at the mature green stage, which showed an increasing trend with the ripening of the fruit, and the increase was large. The content of lycopene in dwarf pink bird reached the highest value, 23.39 mg/kg, during the red and ripening stage. Micro Tom had the second highest at 15.14 mg/kg (Figure 1C). We found that changes in lycopene content were consistent with changes in tomato fruit color.

### 3.2. Differences in Antioxidant Capacity of Tomatoes with Different Fruit Colors

Previous studies have shown the color differences between different varieties of tomato at the red and ripening stage are mainly determined by lycopene content, which has been shown to have significant antioxidant functions [38], and in order to analyze the correlation between the level of lycopene accumulation and antioxidant capacity, the present study determined the activities of the key antioxidant enzymes in tomato fruits at the red and ripening stage. The results of the study showed that there was no significant difference in SOD activity among tomato fruits of different varieties, with dwarf pink bird having the highest SOD activity of 552.64 U·g^−1^FW·h^−1^ (Figure 2A). There was a significant difference in CAT activity among the varieties, with dwarf pink bird having the highest activity of 69.83 U·g^−1^FW·h^−1^, Micro Tom having the next highest, and canary displaying the lowest. This showed that having a high or low content of carotenoids had a significant effect on CAT (Figure 2B). The POD content trend was consistent with that of CAT, and the POD activity of dwarf pink bird was significantly higher than that of other species. Specifically, it was 4.59, 3.26, and 2.21 times that of other species, respectively (Figure 2C).

The relationship between carotenoid content and antioxidant capacity was further verified by the DPPH free radical scavenging assay and the ABTS free radical scavenging assay. Comparing tomato varieties, the speed and ability of scavenging free radicals in dwarf pink bird surpassed those in the rest of the varieties, and the scavenging rate could be as high as 77.3%. Micro Tom scavenged free radicals at a rate and capacity lower than those canaries and red warblers until 1.2 μg/mL, after which canaries had the lowest scavenging rate. Taken together, dwarf pink birds had a better ability to scavenge DPPH, followed by Micro Tom, and the DPPH scavenging rate gradually decreased at a concentration of 2.2 μg/mL of tomato extract (Figure 2D). All tomato samples showed an increasing trend in the scavenging of ABTS radicals. In the concentration range of tomato extract between 0.1 and 0.35 μg/mL, the scavenging rate of Micro Tom was smaller than that of the rest of the species, and there was no significant difference in the scavenging rate of the other species, but the scavenging rates of dwarf pink bird and Micro Tom were greater than those of canary and red warbler when the concentration was more than 0.4 μg/mL, and the scavenging rate of dwarf pink bird and Micro Tom on ABTS radicals at this time could reach more than 70%. Therefore, dwarf pink birds and Micro Tom had better ABTS scavenging abilities, and their scavenging ability also gradually increased with the increasing concentration of tomato extract (Figure 2E). The above results suggest that the variation in tomato fruit color affects the antioxidant capacity of tomato and that therefore there is a link between tomato carotenoids and antioxidant capacity. The above results showed that tomato fruits with more lycopene content had stronger activities in terms of key antioxidant enzymes and free radical scavenging ability, indicating that the content of lycopene in tomato fruits affects the antioxidant capacity of tomato.

### 3.3. Expression Levels of miR394 and Lycopene Metabolic Pathway-Related Enzyme Genes in Different Tomato Varieties

Previous studies have shown that miRNAs can regulate carotenoid synthesis in plants [18,20,21]. The tomato miR394 was obtained in the pre-laboratory stage. To investigate whether SlymiR394 is associated with lycopene synthesis in tomato, we determined the expression of SlymiR394 and the expression of some of the key enzyme genes in the lycopene synthesis pathway. There was no significant difference in the expression of SlymiR394 and key enzymes at the mature green stage, but as the fruit ripened, the SlymiR394 content in canary and red warbler was 2.23 and 1.48 times higher than that in Micro Tom, and the miR394 content in dwarf pink bird was the lowest (Figure 3A). Notably, differences in miR394 expression in tomato fruits of different varieties at the red and ripening stage showed an opposite trend in relation to lycopene content (Figure 1B). The *ZDS* expression of each variety further increased at the red and ripening stage, and the *ZDS* of Pink Bird peaked at the red and ripening stage and was 2.01 times higher than that of Micro Tom (Figure 3B). There was no major difference in PDS gene expression between the mature green stage and turning stage for any variety, but the highest PDS gene expression was found in pink birds at the end of the red and ripening stage. Here, the difference was significant (Figure 3C). In summary, when miR394 expression decreased, the expression of lycopene synthesis pathway-related enzyme genes was elevated, which led to an increase in lycopene content in tomato fruits. Therefore, we hypothesize that tomato miR394 influences the accumulation of lycopene through the regulation of the expression of key enzyme genes of the lycopene synthesis pathway.

### 3.4. Obtaining of STTM394 Transgenic Tomato Plants

To verify the speculation, we used STTM technology to construct an miR394 silencing vector. The stem–loop sequence and miR394 mature sequence were introduced into the pTO2 vector by primer design, and the PCR product of 3250 bp was obtained, which was identified by agarose gel electrophoresis, purified and recovered, digested and ligated, and then ligated into the pFGC5941-Pac I vector. It was screened for clone strains. The positive clones were obtained by sequencing and sequence comparison, and the plant binary expression vectors were successfully constructed as pFGC5941-pOT2-STM394 (Figure 4A). The transgenic plants were obtained by Agrobacterium-mediated plant tissue culture technology. Since the vector used was kanamycin (Kan)-resistant and the explants that failed Agrobacterium infestation were difficult to germinate in the medium with Kan, we screened the resistant tomato plants by adding Kan to the medium used for inducing the generation of healing tissues. DNA was extracted from the screened tomatoes and amplified with specific primers, STTM-New-commen, using wild-type tomato as a negative control. The PCR product was 552 bp, while there was no band in the negative control, which indicated that the STTM394 was successfully transferred into the tomato and transgenic tomato plants were obtained (Figure 4B). Transgenic tomato plants were cultivated to T3 generation, and the expression levels of miR394 and *ZDS* in transgenic tomato plants were detected by qRT-PCR. The relative expression of miR394 in STTM394III-2, STTM394III-3, STTM394III-4 and STTM394III-5 strains during T3 generation was significantly different from that of the wild type, which displayed expressions of 24.7%, 27.8%, 33.6%, and 18.4% (Figure 4C). The relative expression of *ZDS* was 2.36, 2.31, 1.78, and 3.02 times higher than that of WT, respectively (Figure 4D). These results indicated that stable genetic STTM394 transgenic tomato plants had been obtained.

### 3.5. Determination of Lycopene Content in Transgenic Tomato Fruits

Comparing the fruits of wild-type and transgenic tomato plants, there was almost no difference in the color of fruits of WT tomato plants compared to STTM394III-1, STTM394III-2, and STTM394III-3 strains at the mature green stage. At the turning stage, the tomatoes begin to change color and the transgenic tomatoes show an orange-red color. At the red and ripening stage, the fruit color of the transgenic tomato plants shows a bright red color, which is significantly different from that of the wild-type plants (Figure 5A). We calculated the red-to-green ratio per unit area (R_mean/G_mean), and the results were consistent with the color changes we observed, with higher red-to-green ratios in STTM394 transgenic tomatoes at the red and ripening stage compared with wild-type tomatoes, suggesting that the fruits contained a more vibrant red color (Figure 5B). To further compare the color differences between wild-type and transgenic tomato fruits, we determined the lycopene content of the fruits. The lycopene contents of the transgenic plants were all higher than those of the wild type. Among them, the STTM394III-3 strain showed a significant difference with a lycopene content of 26.64 mg/Kg, which was 2.17 times higher than that of the wild type. Secondly, lycopene content in the STTM394III-1 strain also showed upregulation, with a level 1.79 times greater than that of the wild type, and lycopene content in STTM394III-2 strain was only increased by 26.30% compared to WT (Figure 5C). The experimental results showed that the color of STTM394 transgenic tomato fruits showed more vivid red colors at the red and ripening stage, and the silencing of tomato miR394 could increase the lycopene content.

### 3.6. Determination of Antioxidant Capacity of Transgenic Tomato Fruits

To test whether the antioxidant capacity was improved in transgenic tomato fruits, we measured SOD, POD, and CAT activities, as well as DPPH radical and ABTS radical scavenging rates, respectively. The antioxidant enzyme activities of the transgenic tomato plants were significantly higher than those of the wild type, and the SOD activities of STTM394III-1 and STTM394III-3 were 62.49% and 84.16% higher than those of the WT (Figure 6A). Both POD and CAT showed a significant upregulation trend in the transgenic lines, with the highest antioxidant activities of 63.67 U·g^−1^FW·h^−1^ and 67.00 U·g^−1^FW·h^−1^ in STTM394III-3. These values were 3.85 and 1.83 times higher than those of the WT. The POD and CAT activities between the STTM394III-1 and STTM394III-2 strains did not show significant differences, but both were higher than those of the WT (Figure 6B,C). The tomato extracts scavenged both DPPH radicals and ABTS radicals. The scavenging ability of DPPH radicals was better than that of ABTS radicals, and the scavenging rate increased with the increase in concentration. The scavenging rate of DPPH radicals by the transgenic strain was significantly different from that of WT, with scavenging rates of 92.45%, 89.72%, and 85.05%, respectively (Figure 6D). The scavenging of ABTS radicals by the transgenic strains gradually increased with an increasing concentration and converged, with scavenging rates reaching a maximum value of 85% and a minimum value of 79.14% (Figure 6E). The results showed that the antioxidant enzyme activities and the antioxidant capacity were improved in the transgenic tomato fruits of STTM394, suggesting that the increase in lycopene content caused by silencing the tomato miR394 promotes the improvement of antioxidant capacity.

### 3.7. Transgenic Tomato Fruit miR394 Regulates the Expression of Lycopene Biosynthetic Pathway-Related Enzyme Genes

To further investigate how SlymiR394 regulates carotenoid synthesis, we determined the expression of SlymiR394 and the lycopene biosynthetic pathway-related enzyme genes *ZDS* and *PDS* in transgenic tomato fruits. The relative expression of SlymiR394 in transgenic tomato lines decreased to different degrees, showing a significant difference from that in the WT. The expression of miR394 in STTM394III-1 and STTM394III-3 lines was 29.67% and 26.37% of that of the wild type, respectively (Figure 7A). The relative expression of *ZDS* in the lycopene biosynthetic pathway increased significantly in transgenic tomato plants, with values 2.32, 2.19, and 3.58 times that of wild type (Figure 7B). The relative expression of *PDS* was not significantly different from that of the wild type, and the maximum upregulated expression of a single plant was only 1.2 times that of the wild type. Therefore, the silencing of SlymiR394 had no significant effect on the *PDS* content in the metabolic pathway (Figure 7C). In summary, SlymiR394 regulates the expression of enzyme genes related to carotenoid metabolic pathway *ZDS*, thereby increasing the lycopene content and ultimately improving the antioxidant capacity of tomato.

### 3.8. Analysis of SlymiR394 and ZDS Interactions

According to the expression changes in SlymiR394 and important enzymes in the lycopene biosynthetic pathway detected in transgenic tomato varieties, SlymiR394 and *ZDS* showed obvious opposite expression trends (Figure 7A,B). Therefore, we hypothesize that miR394 regulates lycopene accumulation in tomatoes by targeting *ZDS*. Firstly, we analyzed the mature sequences of SlymiR394 and *ZDS* in tomato in the psRNATarget (https://www.zhaolab.org/psRNATarget/, accessed on 24 June 2023) database during raw letter analysis, and the results showed that SlymiR394 might have a targeting relationship with *ZDS* in the lycopene biosynthetic pathway. We also analyzed the action sites. SlymiR394 had base complementarity with *ZDS* and at least 9 complementary bases at the binding site. The mathematical expectation value with *ZDS* was 5.0. The energy value unpaired energy (UPE) was lower than 20. It can be determined from the mode of action that the regulation of miR394 to the target might be cleavage (Figure 8A).

To verify the targeting relationship between SlymiR394 and *ZDS*, we constructed pOT2-miR394 and pTF486-ZDS vectors for subcellular localization experiments (Figure 8B). Using tomato genomic DNA as a template, *ZDS* with a length of 1064 bp containing the binding site and pre-miR394 with a length of 200 bp were cloned from tomato DNA, and the cloned products were recovered, digested, and enzymatically ligated into the vectors pTF486-eGFP and pOT2, respectively, to ultimately obtain pOT2-miR394 (Figure 8C) and pTF486-ZDS vectors (Figure 8D).

The successfully constructed vector was transformed into Agrobacterium and injected into the lower epidermis of tobacco, and the fluorescence intensity of GFP was observed by laser confocal microscopy. The pOT2 + pTF486-GFP combination was used as a blank control to exclude fluorescence interference. pOT2-miR394 + pTF486-GFP and pOT2 + pTF486-GFP had the same fluorescence intensity and were expressed in all parts of the cell. The fluorescence expression of the pTF486-GFP vector linked to *ZDS* was all concentrated in chloroplasts, which is consistent with previous studies showing that *ZDS* genes encoding lycopene accumulate in chloroplasts [39]. pOT2-miR394 + pTF486-ZDS-GFP showed significantly less fluorescence intensity in pOT2-miR394 + pTF486-ZDS-GFP than in pOT2 + pTF486-ZDS-GFP, and the fluorescence intensity was significantly decreased when miR394 coexisted with *ZDS*, suggesting that miR394 could reduce the gene expression of *ZDS* and that miR394 targets *ZDS*. This is consistent with the miRNA-to-target mode of action (Figure 9A). To further determine the interactions between miR394 and *ZDS*, the relative gene expression of GFP in its four cis-transformed tobacco leaves was determined, and the results showed that the GFP content of pOT2-miR394 + pTF486-ZDS-GFP decreased by 27.64% compared with the control group (Figure 9B).

In summary, after psRNATarget software analysis, the subcellular localization and target interactions indicated that miR394 participates in the lycopene biosynthetic pathway and regulates the synthesis and accumulation of lycopene by targeting *ZDS*, which ultimately leads to the enhancement of antioxidant capacity in tomato.

## 4. Discussion

Lycopene is a carotenoid that imparts red color to tomato fruits [40]. Previous studies have shown that lycopene, as an antioxidant, can inhibit lipid oxidation at early stages. It is more effective at this activity compared to other carotenoids [28]. Cells treated with higher concentrations of lycopene (4.0 μg/mL) exhibited significant antioxidant activity, such as SOD, CAT, and GSH-Px activities [41]. This aligns with our findings, which revealed that tomatoes with different fruit colors contain varying levels of lycopene, leading to differences in antioxidant capacity among varieties. This is primarily manifested in antioxidant enzyme activities and free radical scavenging ability. We also found that higher lycopene content correlates with stronger antioxidant capacity in tomatoes, providing new insights for subsequent experiments.

It has been reported that miRNAs can regulate carotenoid synthesis. For example, the Arabidopsis miR156b gene expressed in Brassica napus increased lutein and β-carotene levels in rapeseeds [42]. miRNA regulatory analysis identified miR159a, miR171b, and miR396a as significantly associated with carotenoid biosynthesis in cassava [43]. The overexpression of miR160a in tomatoes enhanced lycopene content [19]. In our preliminary research, we identified a critical miRNA, tomato miR394. Previous studies demonstrated that miR394 plays vital roles in plant growth, development, and stress resistance. miR394 negatively regulates BR signaling via the BIN2-BZR1/BES1 axis to suppress hypocotyl elongation in Arabidopsis [44]. Maize miR394 regulates drought tolerance by targeting the ZmLCR gene [45]. However, whether miR394 regulates lycopene synthesis in tomatoes remains unclear. In our study, we observed that the expression trend of miR394 in tomatoes with different fruit colors aligns with lycopene content. To explore this relationship, we silenced tomato miR394 using STTM technology and generated stable STTM-miR394 transgenic plants via Agrobacterium-mediated tissue culture. We found that silencing miR394 significantly increased lycopene content and upregulated key enzyme genes in the lycopene synthesis pathway. These results confirm that miR394 affects lycopene synthesis by regulating these genes, though the specific targets and mechanisms require further investigation.

MiRNAs primarily regulate gene expression in plants through transcript cleavage or translational inhibition [46]. For instance, miR172 and miR156/7 inhibit the translation of *AP2* and *SPL3*, respectively [47,48], while miR398 and miR165/6 specifically suppress the protein biosynthesis of *CSD2* and *PHB* through target binding [49]. Using the psRNATarget database (https://www.zhaolab.org/psRNATarget/, accessed on 24 June 2023), we predicted interactions between miR394 and its target *ZDS*. Subcellular localization experiments verified that miR394 interacts with *ZDS* in chloroplasts and negatively regulates *ZDS* expression, which is consistent with previous studies. Thus, we conclude that the miR394-ZDS module regulates lycopene accumulation and antioxidant capacity by modulating key enzyme genes in the lycopene synthesis pathway.

In summary, this study measured lycopene content, antioxidant capacity, and expression levels of miR394 and lycopene synthesis-related enzyme genes in different tomato varieties, revealing miR394-associated patterns in lycopene synthesis. By combining STTM technology and Agrobacterium-mediated transformation, we generated stable STTM394 tomato lines and found that silencing miR394 increased lycopene content and enhanced antioxidant capacity. Subsequent psRNATarget-based prediction and subcellular localization experiments confirmed that miR394 regulates lycopene synthesis by targeting *ZDS*. Finally, we elucidated the mechanism of the tomato miR394-ZDS module in regulating lycopene biosynthesis (Figure 10).

This study is the first to clarify the role of tomato miR394 in lycopene metabolism and unveil the regulatory mechanism of the miR394-ZDS module in lycopene synthesis, providing a theoretical and empirical foundation for studying lycopene biosynthesis in tomatoes. However, since experiments were conducted under controlled room conditions, the impact of field environments (e.g., light intensity, temperature fluctuations) on miR394-ZDS regulation remains unassessed. Future studies should conduct multi-ecological field trials to analyze light-temperature interactions on this regulatory module and screen stable, high-yield lycopene-enhanced tomato lines.

## 5. Conclusions

In tomato, miR394 binds to *ZDS* and reduces its expression. Therefore, in STTM394 transgenic plants, silencing miR394 upregulates the expression of its target *ZDS*, thereby promoting lycopene synthesis and conferring enhanced antioxidant capacity to tomatoes.

## Figures and Tables

**Figure 1 biomolecules-15-00857-f001:**
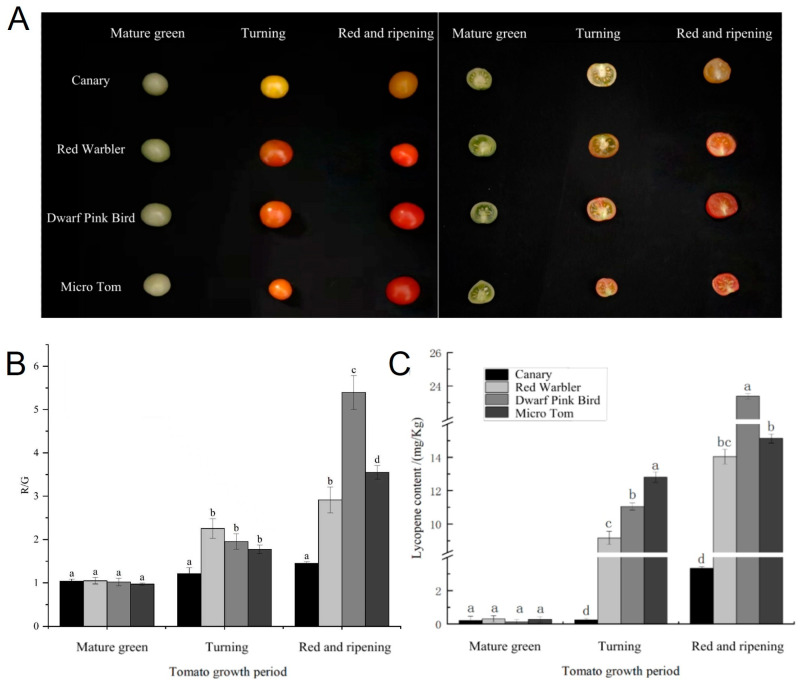
Morphology and lycopene content determination of tomato with different fruit colors: (**A**) fruit color changes in each period of tomato with different fruit colors; (**B**) tomato red–green ratio in each period of tomato with different fruit colors; (**C**) changes in lycopene content in each period of tomato with different fruit colors; different lowercase letters (a, b, c, d) above bars denote these significant differences as determined by one-way ANOVA, *p* < 0.05; groups sharing the same letter are not significantly different. Data are expressed as mean ± standard deviation (*n* = 3).

**Figure 2 biomolecules-15-00857-f002:**
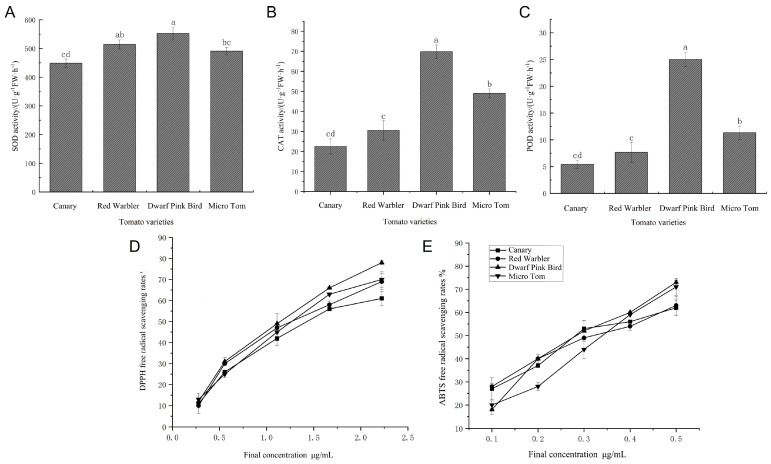
Determination of antioxidant capacity of tomato fruits with different fruit colors: (**A**) measurement of SOD activity of tomato with different fruit colors; (**B**) measurement of CAT activity of tomato with different fruit colors; (**C**) measurement of POD activity of tomato with different fruit colors; (**D**) DPPH radical scavenging rate of different tomato varieties; (**E**) ABTS radical scavenging rate of different tomato varieties. Different lowercase letters (a, b, c, d) above bars denote these significant differences as determined by one-way ANOVA, *p* < 0.05; groups sharing the same letter are not significantly different. Data are expressed as mean ± standard deviation (*n* = 3).

**Figure 3 biomolecules-15-00857-f003:**
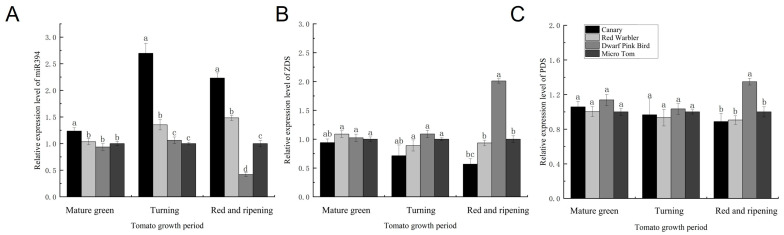
Determination of expression of miR394 and some key enzyme genes during ripening of tomato with different fruit colors: (**A**) miR394 expression in tomato fruits with different fruit colors during each period; (**B**) *ZDS* expression in tomato fruits with different fruit colors during each period; (**C**) PDS expression in tomato fruits with different fruit colors during each period. Different lowercase letters (a, b, c, d) above bars denote these significant differences as determined by one-way ANOVA, *p* < 0.05; groups sharing same letter are not significantly different. Data are expressed as mean ± standard deviation (*n* = 3).

**Figure 4 biomolecules-15-00857-f004:**
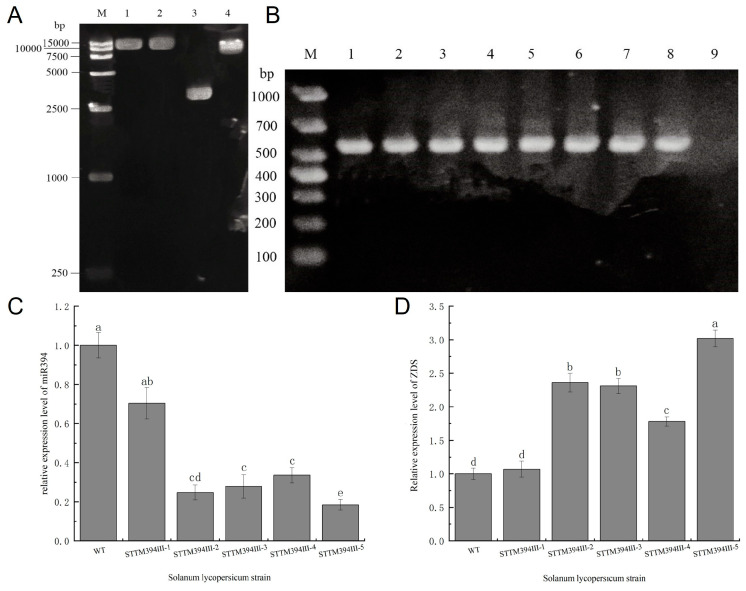
Identification of STTM394 transgenic plants: (**A**) pFGC5941-pOT2-STTM394 vector construction; M: DNA marker (15,000 bp); lanes 1–2 are pFGC5941-STTMmiR394 product bands (11,474 bp); lane 3 is original pOT2-Poly-Cis vector (3705 bp); lane 4 is original pFGC5941 vector (8610 bp). (**B**) PCR identification of transgenic plants; M: DNA marker (1000 bp); lanes 1–8: PCR-amplified bands of transgenic plants (552 bp); lane 9: control of wild-type plants. (**C**) Relative expression assay of miR394 in T3 generation. (**D**) Relative expression assay of *ZDS* in T3 generation. a, b, c, d: significant differences (*p* < 0.05). WT: wild type. STTM394: transgenic lettuce with miR394 silenced by STTM technology. Different lowercase letters (a, b, c, d) above bars denote these significant differences as determined by one-way ANOVA, *p* < 0.05; groups sharing same letter are not significantly different. Data are expressed as mean ± standard deviation (*n* = 3).

**Figure 5 biomolecules-15-00857-f005:**
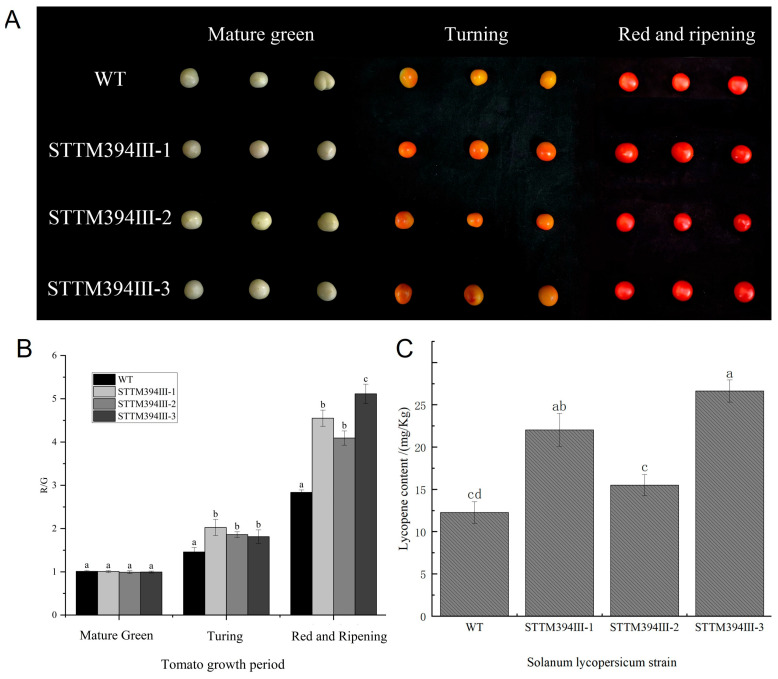
Lycopene content in STTM394 tomato fruits: (**A**) phenotype of STTM394 tomato fruits; (**B**) red–green ratio of STTM394 tomato fruits; (**C**) lycopene content in STTM394 tomato fruits. Different lowercase letters (a, b, c, d) above bars denote these significant differences as determined by one-way ANOVA, *p* < 0.05; groups sharing same letter are not significantly different.

**Figure 6 biomolecules-15-00857-f006:**
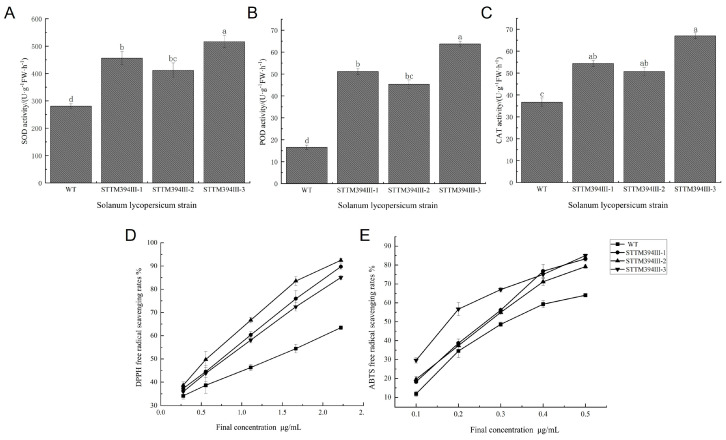
Antioxidant capacity of STTM394 tomato fruit: (**A**) SOD activity assay of STTM394 tomato fruit; (**B**) POD activity assay of STTM394 tomato fruit; (**C**) CAT activity assay of STTM394 tomato fruit; (**D**) DPPH radical scavenging rate of STTM394 tomato fruit; (**E**) ABTS radical scavenging rate of STTM394 tomato fruit. a, b, c, and d: significant differences (*p* < 0.05). WT: wild type. STTM394: transgenic lettuce with miR394 silenced by STTM technology. Different lowercase letters (a, b, c, d) above bars denote these significant differences as determined by one-way ANOVA, *p* < 0.05; groups sharing same letter are not significantly different. Data are expressed as mean ± standard deviation (*n* = 3).

**Figure 7 biomolecules-15-00857-f007:**
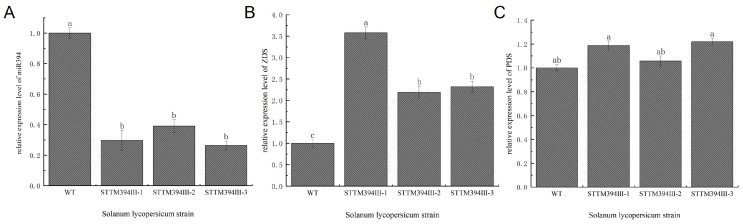
Expression of miR394 and lycopene biosynthetic pathway-related enzyme genes in transgenic tomato fruits: (**A**) expression level of miR394 in STTM394 tomato fruits; (**B**) expression level of *ZDS* in STTM394 tomato fruits; (**C**) expression level of PDS in STTM394 tomato fruits. WT: wild type. STTM394: transgenic lettuce with miR394 silenced by STTM technology. Different lowercase letters (a, b, c) above bars denote these significant differences as determined by one-way ANOVA, *p* < 0.05; groups sharing same letter are not significantly different. Data are presented as mean ± SD (*n* = 3).

**Figure 8 biomolecules-15-00857-f008:**
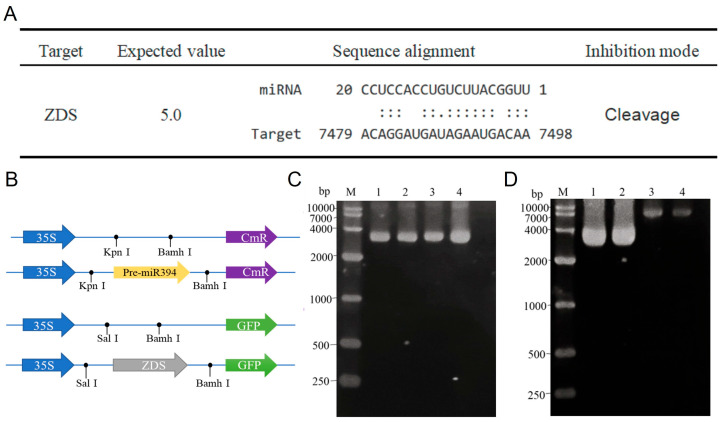
SlymiR394 and *ZDS* target prediction and interaction vector construction: (**A**) MiR394 and *ZDS* interaction site prediction. (**B**) Schematic diagram of interaction vector construction. (**C**) pOT2-miR394 vector construction map and electropherogram. M: DNA marker (10,000 bp). Lanes 1–2 are pOT2-miR394 product bands (3867 bp), and lanes 3–4 are original pOT2 vectors (3705 bp). (**D**) Map of pTF486-ZDS-GFP vector construction and electropherogram. M: DNA marker (10,000 bp). Lanes 1–2 for pTF486-ZDS-GFP product band (5194 bp), and lanes 3–4 for the original pTF486-GFP vector (4148 bp).

**Figure 9 biomolecules-15-00857-f009:**
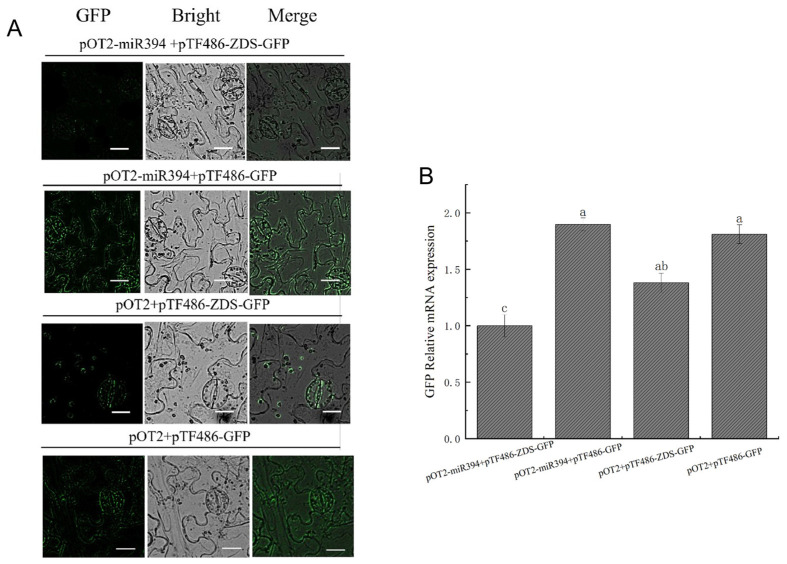
Interaction validation plot of miR394 and *ZDS*. (**A**) Fluorescence validation of interaction between miR394 and *ZDS*. Scale bar: 20 μm. (**B**) Measurement of GFP expression levels. Different lowercase letters (a, b, c) above bars denote these significant differences as determined by one-way ANOVA, *p* < 0.05; groups sharing same letters are not significantly different. Data are presented as mean ± SD (*n* = 3).

**Figure 10 biomolecules-15-00857-f010:**
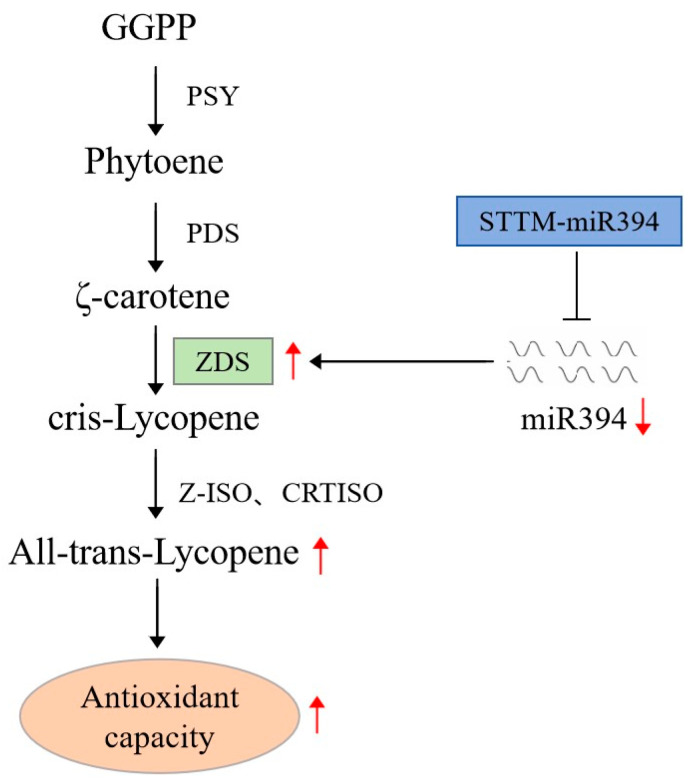
Plot of miR394 acting on *ZDS*. Red arrows indicate up- or down-regulation of expression.

**Table 1 biomolecules-15-00857-t001:** Primer sequence.

Primer Name	Sequence (5′-3′)
miR394-RT	GTTGGCTCTGGTGCAGGGTCCGAGGTATTCGCACCAGAGCCAACGGAGGT
miR394-F	CGCGTTGGCATTCTGTCC
miR394-R	GTGCAGGGTCCGAGGT
U6-F	CATCCGATAAAATTGGAACGA
U6-R	TTTGTGCGTGTCATCCTTGCG
Actin-F	GAAATAGCATAAGATGGCAGACG
Actin-R	ATACCCACCATCACACCAGTAT
PDS-F	CATGCCACGACCAGAAGATTG
PDS-R	CACCAGCAATAACAATCTCCAATG
ZDS-F	TTCATCCATCAACAGGGTACAT
ZDS-R	GCCACAAACCATTCCAAACTC
pTF486-ZDS-GFP-F	ACGCGTCGACGTCGGCCATAGCGGCCGCGGAAGTATTTGAATCAAGTGGCTCCT
pTF486-ZDS-GFP-R	CGGGATCCCGAATTCTTCCTTGTGCGATGC
pOT2-miR394-F	CGGGGTACCAGTAGTGAAGATTATTGAGAGAC
pOT2-miR394-R	CGCGGATCCTCATTATCATCAGTATCAACACA
STTM-miR394-F	GCCATTTAAATATGGTCTAAAGAAGAAGAATTTGGCATTCTctaGTCCACCTCCGAATTCGGTACGCTGAAATCACCAG
STTM-miR394-R	GCCATTTAAATTAGACCATAACAACAACAACGGAGGTGGACAtagGAATGCCAAAAGCTTGGGCTGTCCTCTCCAAATG
STTM-New-Common-PF	CGCACAACCCCACTATCCTT
STTM-New-Common-PR	GGGCCTGCAACCTTATCCTT

## Data Availability

The original contributions presented in this study are included in the article. Further inquiries can be directed to the corresponding authors.

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
