# Peer review of "The SlymiR394-ZDS Module Enhances the Antioxidant Capacity of Tomato by Promoting Lycopene Synthesis"

_biomolecules, 2025, doi:10.3390/biom15060857_

Round 1
Reviewer 1 Report
Comments and Suggestions for Authors
The manuscript focuses on the regulation of carotenoid biosynthesis through specific genes and microRNAs, aiming to shed light on the underlying mechanisms of carotenoid metabolism. While this is a timely and relevant topic, the manuscript suffers from numerous critical weaknesses that undermine its scientific rigor and suitability for publication in Biomolecules. A major concern is the lack of detail regarding the experimental design: the manuscript provides insufficient information on key methodological aspects, such as sample sizes, control groups, experimental protocols, and statistical analyses. These elements are essential for validating findings related to carotenoid biosynthesis and antioxidant capacity. In their absence, the reported results remain largely speculative.
Additional issues further weaken the manuscript’s quality and clarity:
The abstract lacks clarity and conciseness, making it difficult to grasp the study’s main objectives and findings. It also omits a summary of the methodologies employed and fails to include specific quantitative results that substantiate the claims regarding enhanced carotenoid synthesis and antioxidant capacity.
The introduction does not provide a thorough review of relevant literature, citing only 15 references. The rationale for studying miR394 and ZDS in the context of carotenoid biosynthesis is poorly articulated. Furthermore, the introduction does not clearly identify the existing knowledge gaps the study aims to address, nor does it explicitly define the research objectives or hypotheses. There are also numerous issues with scientific terminology and formatting (e.g., "carotenoids... are one of the major phytochemicals" [L.29]; “a lutein carotenoid” [L.40]; Latin names not italicized [L.80–82]).
The Materials and methods section lacks the necessary detail regarding the specific methodologies used; clear protocols, including equipment used and relevant experimental details are essential for replication. The section lacks relevant details on sampling and sample preparation, this information being important for understanding the findings. There is no description of how SOD, POD, and CAT activities were measured (L.136–137), and the reference to “different concentration gradients of standards and extracts” (L.139–140) is vague and lacks an accompanying design description. The sources and quality of reagents are not mentioned. There is no mention of control groups or replicates, these being essential for ensuring the reliability of the results and for statistical analysis. Besides, the section does not outline at all the statistical methods used for data analysis the data.
In #3, some results mention significant differences, but there are no details regarding the statistical analyses performed; including specific p-values, and descriptions of the statistical tests used is a must. The discussion related to color is speculative, since no methodological issues were provided to support it; the reported carotenoid contents is also speculative, since no proper methodological issues were provided to support them (#3.1 and #3.5). #3.2 and #3.6 are also speculative, since no proper methodological issues were provided to support them.
The discussion fails to adequately integrate the results with the broader context of existing research in carotenoid biosynthesis and plant biotechnology. It includes filler content and redundant language, lacks references to prior studies in tomato research, and does not acknowledge any study limitations. Furthermore, it misses the opportunity to propose future research directions or practical applications of the findings.
The manuscript lacks a concluding section. This omission results in an absence of a clear summary of the study’s main findings and their broader implications for future research, agriculture, or human health.
There is no clear statement regarding potential conflicts of interest, which is essential for maintaining transparency and credibility in research.
Given the substantial methodological, structural, and scientific shortcomings outlined above, I cannot recommend this manuscript for publication in Biomolecules.
Comments on the Quality of English Language
The English could be improved to more clearly express the research.
Reviewer 2 Report
Comments and Suggestions for Authors
Dear Authors,
The topic is novel and provides valuable new insights into the regulation of carotenoid production during the ripening of different tomato varieties. However, the manuscript requires several major revisions:
Line 36: The use of the term "them" is unclear and could be misleading. Please rephrase or clarify to avoid confusion.
Line 42: The description of zeaxanthin should be expanded to match the level of detail provided for other carotenoids mentioned earlier.
Line 60: This sentence is unclear and should be rewritten to improve clarity.
Line 79: Define the acronym "amiRNA" upon its first appearance for the benefit of readers who may not be familiar with it.
Line 198: Indicate the specific imaging instrument used to obtain the photos.
Figure 5A: The background in the wild-type "Red and ripening" panel should match the contrast of the panels showing the transgenic tomatoes to ensure visual consistency.
Statistical Analysis: Statistical annotations (e.g., “a”, “b”, “c”) are not explained in the figure legends, and no statistical methods are described in the Materials and Methods section. Clarify how statistical significance was calculated, which tests were used, and against which samples comparisons were made (e.g., in Figure 2).
miR394 Justification: The rationale for selecting miR394 for this study should be clearly stated. Provide background information and include relevant references to support its selection.
Line 289: The explanation of screening with Kan needs to be clearer. Please briefly describe the procedure.
Round 2
Reviewer 2 Report
Comments and Suggestions for Authors
Dear Authors,
The manuscript is now clearer and more fluent, and it has gained greater clarity overall. However, a few aspects still require improvement.
Regarding my previous comment, it is important to mention at least once that KAN is the acronym for Kanamycin.
Moreover, it is still not clear what the different lowercase letters (a, b, c, d) in the statistical analysis refer to. What does "a" represent? What about "b", etc.? Please clarify this in the figure legends or in the text.
